# Impact of Rapid pH Changes on Activated Sludge Process

Kristina Kokina [1,2,*], Linda Mezule [1], Kamila Gruskevica [3], Romans Neilands [3,4], Ksenija Golovko [1] and Talis Juhna [1]

1   Water Research and Environmental Biotechnology Laboratory, Water Systems and Biotechnology Institute, Faculty of Civil Engineering, Riga Technical University, Kipsalas 6b-227, LV-1048 Riga, Latvia; linda.mezule@rtu.lv (L.M.); ksenija.golovko@rtu.lv (K.G.); talis.juhna@rtu.lv (T.J.)
2   Institute of Food Safety, Animal Health and Environment 'BIOR', Lejupes Street 3, LV-1076 Riga, Latvia
3   Department of Water Engineering and Technology, Faculty of Civil Engineering, Riga Technical University, Kipsalas Street 6B-227, LV-1048 Riga, Latvia; kamila.gruskevica@rtu.lv (K.G.); romans.neilands@rtu.lv (R.N.)
4   Riga Water Ltd., Dzintara Street 60, LV-10163 Riga, Latvia
*   Correspondence: kristina.kokina_1@rtu.lv

**Featured Application: Knowledge about inhibition of activated sludge process is crucial for control of the most applied biological wastewater treatment technology.**

**Abstract:** The inhibition effect of rapid variations of pH in wastewater on activated sludge was investigated in laboratory-scale sequencing batch reactors (SBR). The toxic influence of pH 6.5 and 8.5 was examined. The experiment with pH 8.5 was preferable to formation of high FA concentration and showed a low risk of inhibition of second step nitrification (conversion of nitrites to nitrates). However, the reactor at pH 6.5 showed inhibition of first-step nitrification (conversion of ammonia to nitrites) caused by FNA formation. High ammonia levels caused a decrease in the overall microfauna population, whereas low–enhanced *gymnamoebae*, *Zoogloea*, and *Chilodonella* sp. population increased after 72 h of inhibition. Destructive acidic pH influence caused sludge washout from the reactor and, therefore, higher organic load on ASP and intensive sludge foam due to *Zoogloea* higher population.

**Keywords:** activated sludge process; wastewater treatment; inhibiting action; sequencing batch reactors; sludge microfauna population

## 1. Introduction

Activated sludge process (ASP) is the most applied biological wastewater treatment technology involving microorganisms in the removal of pollutants, organic matter, and nitrogen. Nitrogen removal occurs during the process of nitrification by ammonium oxidizing bacteria (AOB) and nitrite oxidizing bacteria (NOB) [1]. This process is often inhibited by specific toxic compounds (pharmaceuticals, hormones, heavy metals) or shock load (rapid increase in the concentration) of some technological parameters [2]. In comparison to nitrification, nitritation is more persistent towards inhibiting action [3]. In wastewater treatment systems, inhibition of NOB is initiated intentionally to establish Anammox process.

Inhibition in the nitrite oxidizing process starts when the free ammonia ($NH_3$ or FA) concentration increases to 0.1–1.0 mg/L and free nitrous acid ($HNO_2$ or FNA) to 0.2–2.8 mg/L; meanwhile, ammonia oxidizing process inhibition occurs when FA concentration is over 10–150 mg/L [4]. In the range of 5–10 mg, FA/L NOB are inhibited but AOB are not affected; simultaneously, nitrite accumulation occurs [5]. The toxicity mechanism of FNA has been observed at a much lower concentration—10 µg/L of the FNA inhibits both aerobic and anoxic cells [6].

Formation of FA and FNA can be caused by pH changes in the wastewater [7]. pH in water and wastewater impacts numerous physicochemical and biological treatment processes, including activated sludge and anaerobic digestion. A rapid increase in pH in the

wastewater treatment plant (WWTP) can be caused by a discharge of industrial wastewater into the sewage systems [8], and a decrease in pH by acidic runoff of salt–sand mixtures applied to roads in winter to avoid skidding [9]. The increase in pH leads to an increase in FA concentration under alkaline conditions (pH > 7). The acidic pH of the wastewater is responsible for FNA production [6]. Nevertheless, alkaline environment has a larger inhibiting action than acidic on activated sludge [10] and the optimal pH for wastewater biological reactors is between 6.5–8.5 [11]. Thus, the rapid changes of pH provoke the appearance of FA and FNA compounds in wastewater, promoting partial inhibition in the nitritation process and drastically affecting the nitrification process.

Sludge microfauna composition changes due to variation in the environment have been identified as a useful indicator for the ASP process quality [2]. The dominance of one or two species, rapid decrease in microfauna diversity, and poor sludge settleability indicate toxicity of the wastewater. At the same time, 2–3 species dominance and high population of *Litonotus, Chilodonella, Vorticella microstoma, Opercularia,* and *Podophrya* may indicate sludge overload [12]. Ammonia nitrogen toxic influence, expressed as *Chilodonella uncinata* and *Acineria uncinata* dominance (50–70%), may also be a sludge overload sign [13].

The aim of this study was to evaluate the effect of rapid pH changes from optimal to alkaline (8.5) or slightly acidic (6.5) on the inhibition-vulnerable nitrification process to assess short-term effects in the sludge. In the present study, controlled pH values were used to find the relationship between pH, total ammonia, and changes in sludge microfauna composition in lab-scale sequencing batch reactors (SBR) reactors.

## 2. Materials and Methods

### 2.1. Experimental Setup

The experimental setup included two lab-scale SBR reactors [14], an automatic controller (Adrona, Riga, Latvia), a peristaltic pump (MasterFlex L/S, Cole-Parmer, Vernon Hills, IL, USA), and air compressors (Marina 50 Air Pump, 50 L/h, Hagen, China) (Schematic diagram S1). The following SBR cycle was used for the study: fill phase (simultaneously with a draw phase) for 25 min, aeration phase for 215 min, settle phase for 30 min (total cycle of 270 min). The reactors were operated at room temperature ($21 \pm 2$ °C).

Regulation of pH = 6.5 was made by using 10 M hydrochloric acid (HCl) and pH = 8.5 by 10 M sodium hydroxide (NaOH).

SBR reactors (hereafter, reactor with pH 6.5—acidic reactor; reactor with pH 8.5—alkaline reactor) operated one control cycle with influent pH = $7.4 \pm 0.1$, $NH_4$-N = $53.0 \pm 7.0$ mg/L and produced up to 96% $NH_4$-N removal. All effluent samples were collected after the end of the acclimatization cycle and at 0 h, 24 h, 48 h, and 72 h after increasing/decreasing the pH level. The amount of influent wastewater was 0.80 L or 60% of working volume (1.33 L) per SBR cycle. Sludge volume indices ($SVI_5$ and $SVI_{30}$) as well as sludge density index ($SDI_{30}$) were measured during the settle phase [2].

Each laboratory-scale SBR test was performed in three replicates (n = 3) with different influent wastewater and inoculum for SBR reactors taken in March and April 2019.

### 2.2. Inoculum for SBR Reactors

The SBRs were fed with wastewater and activated sludge inoculum collected from municipal WWTP in Riga, Latvia (57.02641; 24.00026, PE = 700,000) operating continuous mode plug flow reactors with nitrification and denitrification processes. Under winter conditions, the age of sludge in the WWTP is around 13–15 days, while in summer it is 18–20 days. Hydraulic retention time is 10–14 h, depending on the actual flow rate. Influent wastewater used for the experiment was collected after primary sedimentation, in March–April 2019. The samples were transported to the laboratory within 30 min of sampling, maintaining the environmental conditions.

### 2.3. Water Quality Analyses

HACH protocols and colorimeter (DR/890, HACH, USA) were used for chemical wastewater parameters: total nitrogen (TN)—HACH 10072, ammonium nitrogen ($NH_4$-N)—HACH 10031, nitrate-nitrogen ($NO_3$-N)—HACH 8039, nitrite nitrogen ($NO_2$-N)—HACH 8507, total phosphorus (TP)—HACH 8190, phosphates ($PO_4$)—HACH 8048, chemical oxygen demand (COD)—HACH 8000. The pH, electrical conductivity (EC), temperature, and dissolved oxygen (DO) were controlled using Multi 340i SET B (WTW, Weilheim, Germany).

A high-temperature combustion technique analyzer (FormacsHT, Scalar, Breda, The Netherlands) was used for total carbon (TC), total organic carbon (TOC), and inorganic carbon (IC) measurements based on LVS EN 1484:2000 standard [15].

### 2.4. Activated Sludge Properties

To characterize activated sludge mixed liquor suspended solids (MLSS) [16], sludge volume $SVI_5$; $SVI_{30}$; sludge density $SDI_{30}$ indices; and visual observation of the sludge odor, bulking, floating, and rising after sedimentation were investigated.

To investigate sludge microfauna, the following parameters were analyzed: overall floccule condition, floccule size and density changes, microfauna population, diversity, and species dominance changes. *Ciliate protozoa, rotifer, worm, gymnamoebae, testate amoebae,* and *zoogloea* investigations were carried out according to a previously described methodology [16–18]. The samples were collected during the last 5 min of the aeration phase of the SBR cycle, homogenized by gentle shaking, applied to a glass slide, and calculated using light microscopy (Leica 6000B, Germany). Each sample was analyzed in two repeats.

Sludge microfauna species calculation for 1 mL of the sample was made for Shannon–Weaver diversity index identification [19].

### 2.5. Statistical Analyses

The Shannon–Weaver index was used as a statistical criterion for the characterization of microfauna community changes. Typical values of the Shannon–Weaver index associated with healthy sludge microfauna diversity are between 1.5 and 3.5. The index lower than 1.5 denotes microfauna dominated by one or two species and less diversity [20].

The T-Test (one-tailed distribution; two-sample unequal variance (heteroscedastic)) was used for the determination of changes in parameter significance between reactor with acidic and alkaline. The T-Test result is expressed by *p*-value; if the *p*-value is less than 0.05, the differences between both reactors are significant.

## 3. Results

### 3.1. Changes in Influent and Effluent Wastewater Characteristics

Activated sludge pH assessment was performed with dissolved oxygen (DO) concentration = $8.2 \pm 0.5$ mg/L during the aeration phase. Sludge inoculum mixed liquor suspended solids (MLSS) was $10.2 \pm 2.0$ g/L. MLSS of the SBR working sludge was $2.75 \pm 0.25$ g/L at the start of the experiments. Overall chemical properties of the influent wastewater are summarized in Table 1.

The results showed that the pH level in alkaline reactor was 7.5 right after the acclimatization cycle and then slightly increased to 8.0 at 24 h but did not reach the influent value 8.50 (Figure S1). However, in acidic reactor, pH decreased from 7.4 (at acclimatization cycle) to 6.6 at 24 h. The pH in alkaline reactor was closer to the 'acclimatization cycle' pH = 7.5 due to the nitrification process' tolerance to slightly higher pH than acidic pH, while acidic influence is inappropriate for the nitrification process (ammonium conversion to nitrites).

**Table 1.** Changes in composition of influent wastewater before and after pH regulations (n = 3).

| Parameters | Initial Wastewater | | Acidic Reactor | | Alkaline Reactor | |
|---|---|---|---|---|---|---|
| | Average Value | ±sd | Average Value | ±sd | Average Value | ±sd |
| pH | 7.4 | 0.1 | 6.6 | 0.1 | 8.4 | 0.1 |
| EC | 1417 | 94 | 1593 | 107 | 1571 | 148 |
| T, °C | 21.9 | 1.1 | 22.2 | 0.6 | 22.0 | 1.0 |
| TN, mg/L | 60 | 7 | 57 | 5 | 58 | 8 |
| $NH_4$-N, mg/L | 53 | 7 | 54 | 5 | 55 | 7 |
| $NO_3$-N, mg/L | 5.2 | 0.8 | 3.2 | 0.7 | 3.1 | 1.6 |
| $NO_2$-N, mg/L | 0.020 | 0.004 | 0.016 | 0.003 | 0.007 | 0.001 |
| TP, mg/L | 23.4 | 3.0 | 23.4 | 2.6 | 21.9 | 1.6 |
| $PO_4$-P, mg/L | 19.3 | 2.5 | 18.7 | 3.2 | 17.2 | 2.8 |
| COD, mg/L | 287 | 72 | 249 | 67 | 243 | 81 |
| TOC, mg/L | 54.2 | 18.5 | 64.9 | 8.8 | 66.1 | 13.7 |
| IC, mg/L | 106.4 | 8.0 | 70.2 | 6.7 | 123.9 | 12.0 |
| TC, mg/L | 160.6 | 25.4 | 135.1 | 14.2 | 190.0 | 24.2 |

The differences in IC concentration (Table 1) in the influent of both reactors (IC = 70.2 ± 8.0 mg/L in acidic reactor and IC = 123.9 ± 22.0 mg/L in alkaline reactor) were influenced by pH regulation. IC concentration in the effluent from acidic reactor had a significant decrease compared with control (without inhibitor); IC concentration remained under 3 mg/L from 24 h to 72 h (Figure 1c). The high DO concentration ensured nitrification under aerobic conditions. The pH drop in the acidic reactor affected its alkalinity, which was observed as a decrease in IC concentration. The same trend has been shown earlier [2]. IC concentration in the effluent from alkaline reactor increased after inhibitor addition and reached 36.5 mg/L at 72 h (Figure 1c). Limitation of inorganic carbon source combined with AOB inhibition by alkaline pH can lead to high FA formation and AOB dynamics modifications [21]. Thus, the high IC concentration in alkaline reactor effluent acted as a buffer for the AOB, and the further inhibition process was not intensified. However, in acidic reactor, effluent IC concentration rapidly decreased after the pH regulation and low pH showed a potentially stronger influence on both AOB and NOB, which can also be influenced by the low inorganic carbon source to buffer nitrification process acidity.

Electrical conductivity (EC) increased in both reactor effluents from 0 h to 24 h due to influent EC increase after inhibitor addition. After 24 h, EC stayed almost constant; thus, treated wastewater had higher salinity after inhibitor influence. NaOH and HCl addition contributed to non-biodegradable wastewater fraction increase. Thus, the ASP process had less treatment efficiency because of biological treatment interruption.

COD concentration increased in the acidic reactor effluent between 0 h and 48 h of operation, which is related to sludge washout from the reactor. At 48 h, COD concentration was the highest—170.0 mg/L (COD removal only 30%). However, in the alkaline reactor, effluent COD was under 68.0 mg/L from 0 h to 72 h (Figure S2). The initial COD removal efficiency was 72%; therefore, COD removal was not affected in alkaline reactor (75%).

Total phosphorus (TP) and $PO_4$-P did not change significantly in both reactor effluents from 0 h to 72 h after inhibitor addition (Tables S1 and S2).

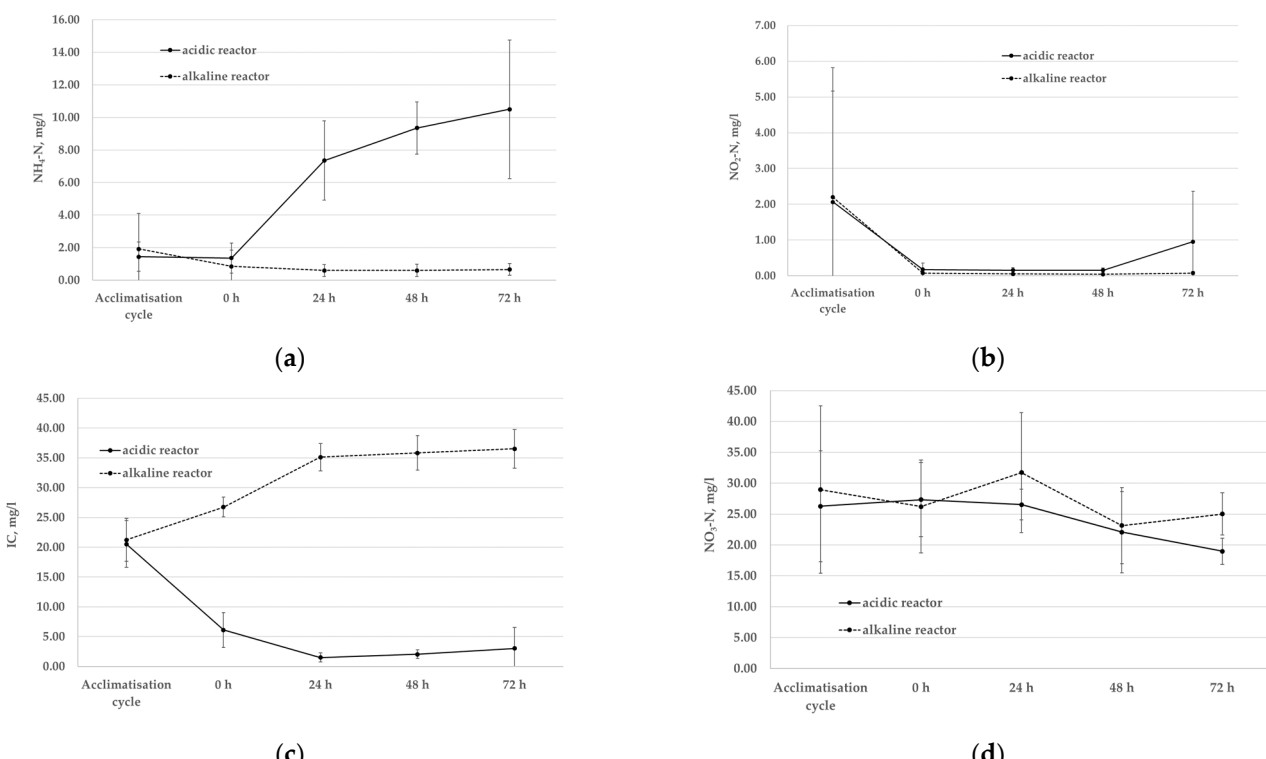

**Figure 1.** Changes of nitrogen and inorganic carbon compounds concentrations in acidic and alkaline pH reactors (n = 3): (**a**) $NH_4$-N; (**b**) $NO_2$-N; (**c**) IC; (**d**) $NO_3$-N.

The results showed that the nitrification process in the reactors before inhibitor addition demonstrated high-quality ammonium nitrogen removal of up to 96–97%. Further, when analyzing the inhibition influence, $NH_4$-N concentration in the effluent from the reactor with alkaline pH stayed up to $1.92 \pm 2.18$ mg/L (Figure 1a). Low $NH_4$-N concentration after 72 h in the alkaline reactor effluent denotes that ammonium oxidizing bacteria (AOB) have not been inhibited, and ammonium was converted to nitrites with the initial efficiency. However, in the acidic reactor, effluent $NH_4$-N concentration increased from $1.44 \pm 0.90$ to $10.50 \pm 4.25$ mg/L within 72 h (Figure 1a).

Nitrite concentration decreased in the effluent of both reactors (Figure 1b) during the 48 h; thus, $NO_2$-N was successfully converted to $NO_3$-N by NOB. Then, an increase in the acidic reactor effluent was observed after 72 h (up to $0.96 \pm 1.41$ mg/L). These changes indicate a possible risk of NOB inhibition due to long FA influence. The nitrate concentration decreased from $26.27 \pm 8.96$ mg/L to $18.98 \pm 2.12$ mg/L in acidic reactor effluent after 72 h and from $28.97 \pm 13.53$ mg/L to $25.03 \pm 3.44$ mg/L in alkaline reactor effluent after 72 h (Figure 1d). As suggested [22], NOB bacteria can be sensitive to the toxicity of free ammonia, and the impact occurs progressively. NOB has a long augmentation time; hence, a slight decrease $NO_3$-N production is observed due to the NOB biomass fraction death after the inhibitor addition and sufficient biomass deficiency for the efficient second-step nitrification process. The formation of high concentrations of the FA leads to strong microbial activity inhibition [22].

It was found that under the alkaline pH conditions, the concentration of FA is 8.82 mg/L (calculated according to Hou et al. [23]), indicating an impact on the nitrification process. In influent wastewater with acidic pH, the predictable formation of FA is low, but the level of FNA reach 0.228 mg/L shows the toxicity towards both AOB and NOB. However, nitrification ratio (NIT) was high (80% in acidic and 96% in alkaline reactors), and nitrite accumulation ratio (NAR) was only 4% in acidic and 0.3% in alkaline reactors (calculated according to Belmonte et al. [24]). Thus, AOB was significantly affected by the inhibition. However, NOB and nitrite accumulation in the sludge was not affected. Statisti-

cal results (T-Test one-tailed distribution; two-sample unequal variance (heteroscedastic)) identified significant changes ($p < 0.05$) in effluent pH, $NH_4$-N, COD, IC in-between acidic and alkaline reactors (Tables S1 and S2).

### 3.2. Changes in Activated Sludge Process

ASP properties changed during the inhibition by acidic and alkaline pH. Visual observation revealed sludge color change (sludge from both reactors became darker). After 24 h of inhibition, constant unpleasant sludge odor from both reactors was detected. However, the odor intensity was stronger in the acidic reactor. The effluent from both reactors became more turbid and the color also changed.

The sludge from the reactor at acidic pH had a dispersed or pin-point floc condition after 48 h. The formation of pin-pointed flocs refers to an environmental shock that keeps sludge microorganisms from agglomeration and causes a formation of a settleable floc. Respectively, pin-point floc condition strongly influences sludge settling abilities [25].

The MLSS at the end of the experiments in acidic reactor was $1.50 \pm 0.30$ g/L and in alkaline reactor $2.25 \pm 0.20$ g/L. Additionally, sludge foaming was observed in the acidic reactor at 72 h of inhibition (Figure S3). White billowy foam usually occurs to young-aged sludge or sludge with low MLSS, sludge with high or too low organic loading, toxic conditions, excessive wasting, and a high abundance of *Zoogloea* spp. [26]. As expected, a rapid decrease in MLSS in the acidic reactor (from $2.75 \pm 0.25$ g/L at 0 h to $1.50 \pm 0.30$ g/L at 72 h) led to an increase in the organic sludge load since the influent wastewater parameters were the same from 0 h to 72 h. Further, pH6.5 increased FNA toxicity risk for ASP, and there was a sludge washout during the experiment (excessive wasting). Besides, *Zoogloea ramigera* and *Zoogloea uva* had considerable augmentation in sludge of the acidic reactor (Figure 2a). Thus, foaming in a reactor with acidic pH denoted ASP troubleshooting due to FNA appearance.

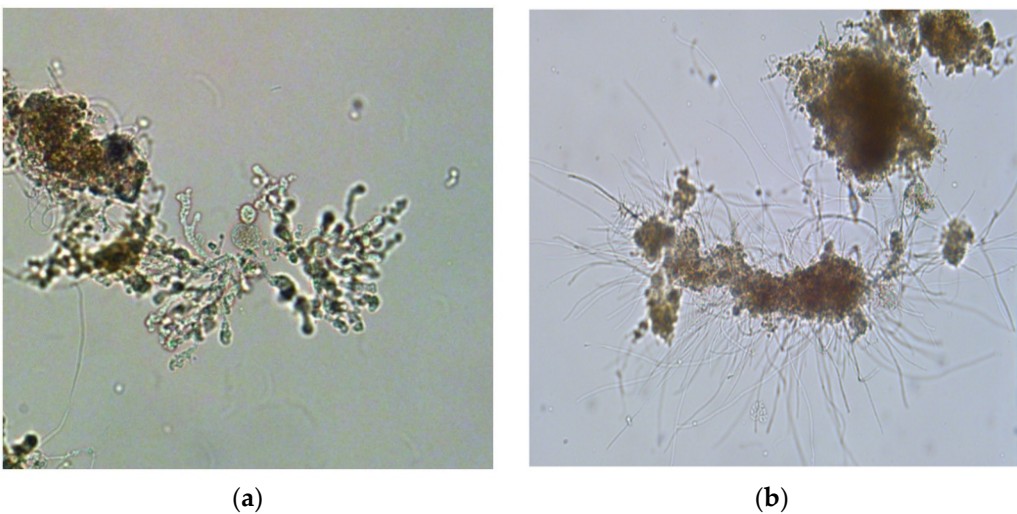

(**a**)  (**b**)

**Figure 2.** *Zoogloea ramigera* (**a**) in sludge of the acidic reactor and filamentous sludge in the alkaline reactor (**b**).

ASP process in the alkaline reactor had filamentous sludge bulking due to an increase in filamentous flocs in the sludge after 48 h (Figure 2b). Filamentous sludge bulking occurs mainly due to the generation of interfloc bridging by filamentous microorganisms, reduction of biomass density, and diffusion of flocs with poor precipitation properties [27]. Figure 2 represents sludge samples from the alkaline after 48 h and a previously reported [28] sample with interfloc bridging by filamentous microorganisms. At the same time, it was observed that total organic carbon (TOC) concentration in the alkaline reactor effluent stabilized and remained at 15.4 mg/L from 0 h to 72 h. TOC concentration in acidic

reactor effluent increased gradually from 15.6 mg/L at 0 h to 21.0 mg/L at 72 h; so, in this reactor, ASP process organic carbon biodegradation was also affected by acidic pH 6.5.

Sludge volume indices ($SVI_5$ and $SVI_{30}$) increased in both reactors after the start of inhibition by the acidic and alkaline pH. $SVI_5$ increased from 222 mL/g to 256 mL/g after 48 h and decreased again to 230 mL/g in acidic reactor (Figure S4). $SVI_{30}$ index in the alkaline reactor was not strongly affected by high pH (Figure S5). The $SVI_{30}$ index had values up to 90 mL/g, which indicate proper sludge settleability despite the fact of increased filamentous sludge flocs in the reactor. However, sometimes, filamentous sludge bulking can occur at relatively low SVI values [25]. Hence, the $SVI_{30}$ index is not the main indicator of the formation of high FA concentration in this case. Although, $SVI_5$ changed more than $SVI_{30}$ and can be specified as the first sign of the inhibition due to an increase from 225 mL/g (acclimatization cycle) to 238 mL/g at 0 h. The risk of the sludge bulking due to high $SVI_{30}$ can be at SVI higher than 120 mL/g [27], which might occur after 72 h of inhibition (Figure S5) due to the gradual increase in $SVI_{30}$ in acidic reactor.

### 3.3. Sludge Microfauna Changes

According to the Shannon–Weaver index, sludge microfauna initial diversity (Table 2) was high and demonstrated even distribution between species. The diversity index stayed at a satisfactory level between 1.5 and 3.5 throughout the tests. This can be characterized as the retaining of sludge microfauna richness without degradation and no formation of specific dominating species. Nevertheless, certain differences between initial and ultimate microfauna species abundance in the acidic and alkaline reactors were observed. The sludge microfauna population in acidic reactor remained at a similar level as the initial sludge microfauna (acclimatization cycle—2566 ind/mL and at 72 h—2427 ind/mL). However, the alkaline reactor sludge microfauna population decreased ('acclimatization cycle'—2566 ind/mL, at 72 h—1716 ind/mL), with the main decrease occurring between 24 h and 48 h (Table 2).

According to the results, the Shannon–Weaver index was still in the range of 2.33–2.71. The alkaline reactor in general did not demonstrate a significant increase in any specific species. Some increase in *Epistylis* and *Tokophrya* was observed, indicating the presence of purification process; at the same time, a decrease in the crawling ciliates can indicate a deterioration in process quality. In general, a specific dominating microfauna indicator was not observed in reactor with alkaline conditions. However, there was a noticeable increase in *gymnamoebae* (*Amoeba limax* and *Mayorella* spp.—from 23 ind/mL to 123 ind/mL and from 103 ind/mL to 343 ind/mL, respectively), *Zoogloea* spp. (*Z. ramigera* and *Z. uva*—from 60 ind/mL to 383 ind/mL and from 50 ind/mL to 190 ind/mL), and *crawling ciliate Chilodonella* sp. (from 27 ind/mL to 187 ind/mL) in the acidic reactor. Possibly, slightly acidic conditions are preferable to *gymnamobae*, *Zoogloea* spp., and *crawling ciliate Chilodonella* sp. development. *Gymnamoebae* (a group of *A. limax* and *Mayorella* spp.) are indicators of very high load. At the same time, *Chilodonella* sp. is an indicator of good effluent quality and low oxygen content [17]. *Zoogloea* spp. development occurs at a high F/M ratio, with increase in FNA concentration or low oxygen. *Z. ramigera* is a floc-former, which interferes with sludge settleability and sludge density similar to filamentous bulking [29]. Consequently, pH decrease enhanced the increase in volatile organic acid concentration (as reported, volatile fatty acids (VFA) concentration and yield are the highest at pH 6 [30]), which led to *Zoogloea* spp. development. Thus, sludge settleability is influenced and washout from the reactor occurs, consequently causing sludge foaming due to an increased F/M ratio after sludge washout.

**Table 2.** Microfauna changes (n = 3) in the acidic and alkaline reactors.

| Average (n = 3) | | Acclimatization Cycle | Acidic Reactor, ind/mL | | | | Alkaline Reactor, ind/mL | | | |
|---|---|---|---|---|---|---|---|---|---|---|
| **Microfauna Group** | **Genus/Species** | | **0 h** | **24 h** | **48 h** | **72 h** | **0 h** | **24 h** | **48 h** | **72 h** |
| gymnamoebae | *Amoeba limax* | 23 | 33 | 90 | 80 | 123 | 37 | 37 | 57 | 53 |
| | *Mayorella* | 103 | 160 | 310 | 330 | 343 | 197 | 303 | 173 | 267 |
| testate amoebae | *Arcella* | 117 | 103 | 90 | 90 | 57 | 157 | 90 | 73 | 60 |
| crawling ciliates | *Chilodonella* | 27 | 43 | 40 | 57 | 187 | 50 | 40 | 123 | 33 |
| | *Aspidisca* | 140 | 120 | 63 | 93 | 47 | 140 | 57 | 37 | 37 |
| free swimming ciliates | *Litonotus* | 330 | 583 | 250 | 60 | 37 | 493 | 247 | 73 | 77 |
| | *Prorodon* | 537 | 567 | 550 | 420 | 217 | 583 | 500 | 287 | 220 |
| | *Holophrya* | 60 | 30 | 57 | 57 | 23 | 23 | 47 | 60 | 40 |
| | *Glaucoma* | 50 | 27 | 33 | 77 | 87 | 40 | 43 | 77 | 60 |
| | *Spirostomum* | 20 | 37 | 13 | 17 | 3 | 17 | 33 | 17 | 13 |
| stalked ciliates | *Epistylis* | 436 | 465 | 348 | 156 | 440 | 239 | 396 | 251 | 436 |
| | *Carchesium* | 77 | 30 | 50 | 40 | 20 | 13 | 33 | 47 | 23 |
| | *V. convallaria* | 93 | 80 | 60 | 73 | 73 | 67 | 133 | 50 | 50 |
| | *V. microstoma* | 347 | 150 | 193 | 140 | 127 | 187 | 163 | 57 | 30 |
| carnivorous ciliates | *Acineta* | 0 | 0 | 0 | 0 | 0 | 0 | 0 | 3 | 0 |
| | *Tokophrya* | 3 | 0 | 0 | 3 | 3 | 3 | 3 | 10 | 17 |
| rotifers | *Rotaria* | 47 | 40 | 40 | 37 | 37 | 63 | 60 | 50 | 60 |
| | *Cephalodella* | 20 | 30 | 27 | 17 | 10 | 30 | 27 | 20 | 27 |
| worms | *Nematoda* | 23 | 20 | 17 | 20 | 7 | 7 | 3 | 10 | 3 |
| | *Aeolosoma* | 3 | 3 | 10 | 13 | 13 | 7 | 13 | 17 | 10 |
| zoogloea | *Z. ramigera* | 60 | 70 | 163 | 293 | 383 | 73 | 100 | 150 | 130 |
| | *Z. uva* | 50 | 47 | 87 | 120 | 190 | 43 | 67 | 93 | 70 |
| Total abundance, ind/mL | | 2566 | 2639 | 2492 | 2193 | 2427 | 2469 | 2396 | 1734 | 1716 |
| Shannon–Weaver index | | 2.44 | 2.33 | 2.50 | 2.60 | 2.48 | 2.40 | 2.50 | 2.71 | 2.49 |

## 4. Discussion

The unionized form of ammonium—free ammonia ($NH_3$ or FA) and free nitrous acid ($HNO_2$ or FNA) in the wastewater [7] are directly influenced by wastewater pH. An increase in pH leads to an increase in FA concentration. Under alkaline conditions (pH > 7), the concentration of FNA will increase as the pH decreases. The principle is characterized by ionized ammonium ($NH_4^+$) and free ammonia ($NH_3$) equilibrium (1) and the nitrous acid equilibrium (2)

$$NH_3 + H_2O \leftrightarrow NH_4^+ + OH^- \tag{1}$$

$$H^+ + NO_2^- \leftrightarrow HNO_2 \tag{2}$$

The concentrations of $NH_3$ and $HNO_2$ are a function of total ammoniacal nitrogen (TAN = $NH_4^+$ + $NH_3$) and total nitrite concentrations ($NO_2^-$ + $HNO_2$), pH, and temperature [4]. Inorganic carbon source is necessary for the alkalinity requirements to buffer the acidification of the sludge due to the nitritation process [2].

To evaluate the effect of rapid pH variation on the activated sludge process, two critical pH levels, 6.5 and 8.5, were selected for lab-scale tests under the study. Calculated concentration of FNA in acidic reactor was 0.228 mg/L ($NH_4$-N = 53.0 ± 7.0 mg/L, T = 20 °C, pH 6.5) and concentration of FA in alkaline reactor was 8.82 mg/L ($NH_4$-N = 53.0 ± 7.0 mg/L,

T = 20 °C, pH 8.5). These concentrations are high enough to cause a toxic effect on the ASP [5,6].

Typical municipal wastewater has approximately 30 mg/L of total nitrogen (TN), temperature = +20 °C, and pH = 7–8 and contains 0.14–1.38 mg/L FA [31]. A further increase in pH and TN in municipal wastewaters could be toxic for NOB due to the formation of excess FA. An inhibition of AOB can be enhanced by the limitation of inorganic carbon (IC) source for several days [21]. Hence, simultaneous limitation in IC source and an increase in FA concentration can lead to destruction of the first-step nitrification and, consequently, to the nitrogen removal process failure.

pH is a technologically important parameter in the biological wastewater treatment process, especially in nitrification and denitrification efficiency. pH 6.5 is more destructive for the activated sludge process than pH 8.5.

Generally, the increase in ammonia concentration is caused by AOB inhibition in the activated sludge in acidic reactor (Figure 1a). The same trend has been shown earlier [2]. Additionally, sludge foaming was observed in this reactor at 72 h of inhibition (Figure S3). Therefore, pH 6.5 is not suitable for the nitrification process. Experiments with alkaline pH were associated with the formation of high FA concentration and showed a potential risk of inhibition of second-step nitrification (conversion of nitrites to nitrates). However, the acidic reactor showed strong inhibition of AOB, probably caused by FNA formation. High ammonia levels and a decrease in pH leads to *zoogloea*, *gymnamoebae*, and *Chilodonella* sp. development; this affects sludge settleability, and sludge washout from low FA reactor occurred. Hence, the organic load increased after sludge washout, and sludge foaming was observed. Increase in pH caused in filamentous sludge bulking was due to filamentous microorganisms' high pH preference. The differences in IC concentration in the influent of both reactors contributed by pH regulation have a significant effect on biomass. The high IC concentration in effluent acted as a buffer for the AOB, and the further inhibition process was not intensified. The negative influence on both AOB under acidic pH can be explained by the low inorganic carbon source to buffer nitrification process acidity. Results showed that nitrification (NIT) was high and nitrite accumulation ratio was low (NAR) in both reactors.

As reported, nitrification is the most efficient in a narrow pH range of 7.8–8.0 [32]. Acidic pH enhances free nitrous acid (FNA) production, which is considered highly toxic for the biological nitrogen removal process. Even 0.0013 mg $HNO_2$/L can inhibit AOB bacteria [33]. Consequently, AOB and NOB bacteria inhibition with acidic pH have different outcomes when compared with alkaline pH.

pH, temperature, and ammonium are useful indicators of a good nitrification process. These parameters can be used as quality parameters for monitoring inhibition of bioreactors and in online early warning systems to reduce potential failures in a WWTP.

## 5. Conclusions

Rapid changes in pH to acidic or alkaline level that are still in the optimal boundary level of wastewater have different effects on ASP:

1. Rapid changes of pH from optimal to acidic had a greater impact on the ASP than alkaline pH.
2. Acidic pH affected ammonia conversion to nitrites indicating inhibition of both AOB and NOB; however, this phenomenon may also be attributed to the limitation of inorganic carbon source (due to pH change itself).
3. Organic carbon biodegradation also decreased, and increased sludge foaming and sludge washout were noted in acidic pH reactor.
4. The slight changes in sludge microfauna population were observed in both reactors. In the acidic reactor, there was a noticeable increase in *gymnamoebae*, *Zoogloea* spp., and *crawling ciliate Chilodonella* sp. In the alkaline pH reactor, the microfauna population decreased and filamentous sludge bulking occurred due to the increase in filamentous flocs.

5.  SVI$_{30}$ index is not the main indicator of the formation of high FA concentration in this case. Although, SVI$_5$ changed more than SVI$_{30}$ and can be specified as the first sign of the inhibition.

**Supplementary Materials:** The following supporting information can be downloaded at https://www.mdpi.com/article/10.3390/app12115754/s1, Figure S1: Changes in pH in acidic and alkaline pH reactors (n = 3); Figure S2: Changes in COD concentration in acidic and alkaline pH reactors (n = 3); Table S1: T-test one tailed results for acidic and alkaline reactor comparison; Table S2: two-sample unequal variance for acidic and alkaline reactor comparison; Figure S3: Sludge foaming in acidic pH lab-scale reactor; Figure S4: Changes in SVI5 in acidic and alkaline pH reactors (n = 3); Figure S5: Changes in SVI30 in acidic and alkaline pH reactors (n = 3); Figure S6: The schematic diagram of experimental set-up [14].

**Author Contributions:** Conceptualization, K.K. and R.N.; methodology, K.K.; validation, K.G. (Ksenija Golovko), K.G. (Kamila Gruskevica), and R.N.; formal analysis, K.G. (Ksenija Golovko); investigation, K.G. (Ksenija Golovko) and K.K.; resources, R.N.; data curation, K.G. (Ksenija Golovko); writing—original draft preparation, K.G. (Ksenija Golovko); writing—review and editing, L.M.; visualization, K.G. (Kamila Gruskevica); supervision, T.J.; project administration, L.M.; funding acquisition, L.M. All authors have read and agreed to the published version of the manuscript.

**Funding:** This research was funded by ERDF Project "Zero-to-low-waste technology for simultaneous production of liquid biofuel and biogas from biomass", No. 1.1.1.1/18/A075.

**Institutional Review Board Statement:** Not applicable.

**Informed Consent Statement:** Not applicable.

**Data Availability Statement:** Not applicable.

**Acknowledgments:** We thank Riga Water Ltd. for providing test materials and supported microbial testing. We thank Jurijs Resetilovs from RESETILOVS Ltd. for additional technological consultations.

**Conflicts of Interest:** The funders had no role in the design of the study; in the collection, analyses, or interpretation of data; in the writing of the manuscript, or in the decision to publish the results.

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
