# Peer review of "Impact of Rapid pH Changes on Activated Sludge Process"

_applsci, doi:10.3390/app12115754_

Round 1

Reviewer 1 Report

Title: Impact of rapid pH changes on activated sludge process

This manuscript entitled “Impact of rapid pH changes on activated sludge process”. This proposition is meaningful. In general, the water quality detection is relatively comprehensive and the analysis of results is in detailed. However, the experimental design is too simple and the discussion is not deep enough. Some ideas(causal relationship or mechanism) are not clearly explained. Its content is based on phenomenological statements. Moreover, there are some writing errors in this article. Therefore, I think the manuscript can be published on Applied Sciences after major revision. The detailed opinions are as follows:

1. In line 44-46, this sentence is confusing. Could you please explain?

2. The experimental design is simple. Is DO=7.5-7.6 representative in this experiment? Under aerobic conditions, the difference of DO would affect the nitrification process.

3. In line 153-154, please elaborate the dynamics modifications of AOB. In line 158, “low pH showed a potentially stronger influence on AOB and NOB”, why AOB would be inhibited by FA under IC limitation conditions? Is this a conjecture or a statement? What is the relationship between IC, FNA and FA? Please provide evidence to support it.

4. In line 166-167, “COD concentration increased in……sludge washout from the reactor”, the COD in sludge is worth pondering. Please supplement relevant data. Too low or too high pH would affect the determination of COD. In line 169-170, the conclusion is questionable.

5. In line 174, the subscript of NH4+-N is wrong. In line 183, the subscript of NO2--N and NO3--N is wrong. In line 178, the formal of “72 h” is wrong. In Figure 1c, the legend of alkaline reactor is missing.

6. In line 177, why NH4+-N concentration increased in the acidic reactor? Please supplement and explain.

7. The discussion is lack of the changes on phosphorus concentration. Please supplement.

8. In line 265-266, “ The alkaline rector did not demonstrate a significant increase in any species”, but it was obvious that many species decreased. Would you please analyze and speculate this phenomenon? The analysis and discussion in this part are not comprehensive enough.

9. In discussion, it is superficial statement based on result and phenomenon and it doesn’t rise to a deeper level of cause analysis or mechanism discovery. For example, why rapid pH variation could affect the change of a series of indicators? Could you please explain it from the level of acid-base theory? Whether chemical bond breaking and bonding occur? The change of IC source is affected by rapid pH variation? AOB were inhibited by FA or FNA. Under IC limitation conditions, is these inhibitions were higher? Could you please state these causality or mechanism?

10. In Figure S3, how do you control the temperature in your reactor? Please explain.

Author Response

We thanks Reviewer 1 for comments and questions. Please find response in attachment.

Reviewer 2 Report

Kokina et al presented a manuscript entitled: “Impact of rapid pH changes on activated sludge process”.

Overall, the manuscript fits the scope of Applied Sciences. Here are my comments/suggestions to be added:

1- Experimental setup: A scheme is needed for this part, so the reader can imagine the experimental work.

2- The authors could add a major reaction scheme for each tested pH 6.5 and 8.5, as per the conclusion point 4.  

3- Generally speaking, how the industry can benefit from this study for real ASPs?  

Author Response

We thanks Reviewer 2 for comments and questions. Please find response in attachment.

Reviewer 3 Report

This manuscript does an excellent research work demonstrating the impact of rapid pH changes on the activated sludge process. I think this paper will be handy and demanded by the journal's readers.
Apart from one comment regarding Fig. 1(c), where the designation of the alkaline reactor line is missing, no more errors or typos were found.

Author Response

We thanks Reviewer 3 for comments. Please find response in attachment.

Round 2

Reviewer 1 Report

The authors have made substantial revisions according to the reviewers comments. It can be published as it is.